# FIAT: FUSING LEARNING PARADIGMS WITH INSTRUCTION-ACCELERATED TUNING

## ABSTRACT

Learning paradigms for large language models (LLMs) currently tend to fall within either in-context learning (ICL) or full fine-tuning. Each of these comes with their own trade-offs based on available data, model size, compute cost, ease-of-use, and final quality with neither solution performing well across-the-board. In this article, we first describe ICL and fine-tuning paradigms in a way that highlights their natural connections. Based on these connections, we propose a new learning paradigm called FIAT[1] that fuses[2] the best of these paradigms together, enabling prompt-engineered instructions and chain-of-thought reasoning with the very *largest models* while also using similar methods to perform parameter updates on a *modestly-sized LLM* with parameter-efficient tuning. We evaluate FIAT's effectiveness on a variety of multilingual tasks[3] and observe that FIAT performs better than both ICL and fine-tuning at scales ranging from 100–10,000 training examples. We hope that FIAT provides a practical way of harnessing the full potential of LLMs without needing to make a hard choice between learning paradigms.

## 1 INTRODUCTION

Large language models (LLMs) show impressive generalization ability to new tasks and languages. Some of their most exciting capabilities, such as producing logical reasoning to solve a problem, are found to emerge only when the model size is over a certain threshold, often hundreds of billions of parameters (Wei et al., 2022b;a). The impressive capabilities of these models to produce high-quality responses without any task-specific tuning along with the very high cost of further tuning such models has led much recent work to focus on the paradigm of In-Context Learning (ICL)—placing a few task-specific examples and instructions into the model's input (Brown et al., 2020; Chowdhery et al., 2022; Google et al., 2023; OpenAI, 2023).

Although prior work has seen that fine-tuning a model on task data can often lead to superior performance on the downstream task compared to ICL (Scao & Rush, 2021; Schick & Schütze, 2020a;b; Asai et al., 2023), there are significantly fewer recent efforts on fine-tuning models for tasks with limited data, perhaps because the time and compute costs associated with tuning a very large model drives practitioners toward smaller models, abandoning the ability to take advantage of emergent model capabilities.

ICL and model fine-tuning each come with their own trade-offs. ICL does not incur any training cost and it allows one to utilize the most capable LLMs (Schick & Schütze, 2020b; OpenAI, 2023). However, while ICL can achieve competitive performance on many tasks with a handful of annotated examplars, it often requires very large models to work well and it cannot take advantage of additional training examples if they do not fit into the context window. For many tasks, this leads to ignoring a substantial amount of potentially-useful training examples. Fine-tuning, on the other hand, is not constrained by the need to fit training examples into the model's input, and it can be quite effective even with smaller language models. These trade-offs tend to lead practitioners to arbitrarily pick a paradigm or run costly experiments on these disparate methods in order to choose the best approach.

---

[1]We derive the name FIAT from *F*using Learning Paradigms with *I*nstruction *A*ccelerated *T*uning.

[2]FIAT fuses not only the learning paradigms but the models themselves.

[3]We say that these tasks are *naturally* low-data because no additional data is available for such languages and it's non-trivial to obtain more; we contrast this with artificially low-data scenarios where large data exists, but is ignored.

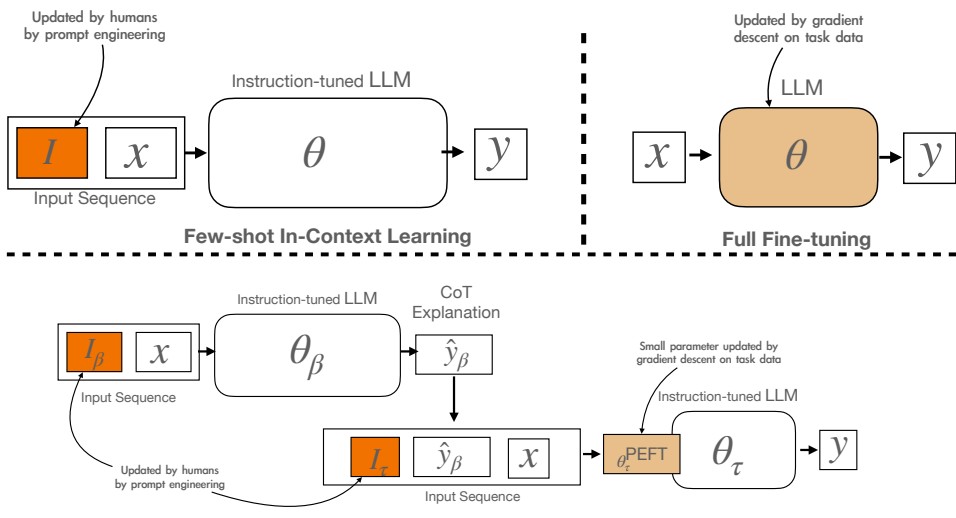

Figure 1: Overall flow of FIAT and how it compares to ICL and fine-tuning. The colored components are updated while building and learning a task-specific instance of FIAT, while other components are fixed. $\theta_\beta$ is the parameters of the larger LLM and $I_\beta$ are the instructions used to induce reasoning; $\theta_\tau$ are the parameters of a moderately-sized LLM to be tuned and $I_\tau$ is its instructions, which helps the model predict the correct final answer.

We instead take the view that these two model learning paradigms are in fact complementary. To this end, we propose FIAT—Fusing Learning Paradigms with Instruction-Accelerated Tuning (FIAT), which utilizes both ICL on very large models and parameter tuning on moderately-sized LLM while fusing the common techniques associated with each paradigm. FIAT uses hand-engineering instruction prompts that elicit chain-of-thought reasoning from a very large model, while also using the generated reasoning and instruction prompts to tune a moderately-size LLM with parameter-efficient tuning. Figure 1 shows the workflow of FIAT and how it compares to ICL and fine-tuning.

In the remainder of this article, we formally describe the connections between ICL and fine-tuning, along with the various techniques that have developed within each paradigm (§2); we propose FIAT, which fuses the best of these together and avoids many of the pitfalls of each of the individuals (§2.3); we present experiments demonstrating how FIAT improves over both learning paradigms in data scenarios ranging from 100–10,000 examples along with ablations detailing where these gains come from (§3).

## 2 LEARNING PARADIGMS FOR LLMS

In this section, we review two popular learning paradigms for LLMs (ICL in §2.1 and parameter tuning in §2.2) while considering their strengths and weaknesses, which directly lead to FIAT (§2.3).

### 2.1 IN-CONTEXT LEARNING

**Instructed ICL** keeps the parameters of the LLM fixed, but it instead selects an instruction prompt (often through manual optimization) to improve the accuracy of the downstream task. Formally, a model prediction is made by sampling[4] a very large pre-trained LLM parameterized by fixed $\theta$ and a textual instruction $I$:

$$P(y|x; \theta, I) \tag{1}$$

While the instructions $I$ are prefixed onto the model input $x$ in practice, we intentionally notate them as an argument of the model, which we argue better reflects how they are conceptualized; we will build on this later.

---

[4]Typically, the sampling is a simple `argmax` with temperature 0, though this isn't always the case as in techniques such as majority voting.

**Chain-of-thought reasoning**    pushes instructed ICL a step further by crafting $I$ to induce step-by-step *reasoning* in the output of the model that improves the model's ability to arrive at a correct prediction (Wei et al., 2022b). This allows auto-regressive inference to output observations about the input or solve sub-problems of the overall task that future decoding steps can leverage when predicting the final answer; it may also elicit textual patterns that the model saw during pre-training, that would otherwise be difficult to access in the model's latent feature space (e.g. via fine-tuning).

**Few-shot ICL**    Few-shot ICL differs from instructed ICL in that its instructions $I$ are composed of a small number of examplars selected among training examples $\mathcal{D}$ that have been formatted as a textual input to the model via instructions.

**Instruction-tuned Base Models**    Instruction-tuned models such as FLAN and T0 (Sanh et al., 2021; Chung et al., 2022; Longpre et al., 2023) often provide significant improvements on ICL compared to using a pre-trained model. This is because instruction-tuning is essentially a second stage pretraining using a set of multitask data whose distribution is closer to the downstream task.

The ICL paradigm achieves competitive results on various tasks with no or only a handful of annotated examples. While it does not incur any additional model tuning cost, ICL often has high inference cost because it requires LLMs over a certain size to work well, especially when using techniques such as chain-of-thought. It also cannot take advantage of additional task data beyond what fits into the context window of the model.

## 2.2    PARAMETER TUNING

**Full-Parameter Fine-tuning**    Given pre-trained parameters $\theta$ of a LLM to tune,[5] standard fine-tuning simply optimizes all parameters of the model on task-specific supervised training data $\mathcal{D}$ according to:

$$P(y|x; \theta) \tag{2}$$

The optimization of $\theta$ is similar in purpose to the process of human prompt engineering of $I$ in ICL.

Since model fine-tuning does not have to fit training data into the context window of the model, it is more effective when there are slightly more training examples available. Fine-tuning also works well on smaller language models with enough training examples, leading to faster inference. However, fine-tuning incurs additional training cost and requires access to model parameters, while some of the most capable LLMs are available for inference-only API access. The model could also easily overfit to the training examples due to catastrophic forgetting (Goodfellow et al., 2013), especially for tasks with limited data.

**Parameter-efficient Fine Tuning**    (PEFT) improves the tuning procedure by using a learning parameterization $\theta^{\text{PEFT}}$ where $|\theta^{\text{PEFT}}| \ll |\theta|$. Besides reducing the danger of overfitting, this learning technique also avoids forgetting features that may be useful for generalization beyond the training set. Similarly, ICL avoids catastrophic forgetting by only modifying the input to the model while keeping the parameters fixed.

## 2.3    FUSING LEARNING PARADIGMS WITH FIAT

In this section, we construct FIAT, motivating the purpose of each design choice in terms of modeling capabilities. ICL and fine-tuning each have compelling strengths along with pitfalls, which we summarize in Table 1. At a high level, we observe that these properties are largely *complementary*.

---

[5]In practice, $|\theta|$ tends to be much smaller for fine-tuning than for ICL.

|  | ICL | Fine-tuning |
|---|---|---|
| *Strengths* | | |
| Works well with small model | No | Yes |
| Supports large training data | No | Yes |
| Supports chain-of-thought reasoning | Yes | No |
| Usage of instruction prompts | Yes | No |
| *Challenges* | | |
| No parameter updates | Yes | No |
| Avoids catastrophic forgetting | Yes | No |

Table 1: Comparison of the ICL and fine-tuning learning paradigms, according to common usage patterns.

Reflecting on these abilities of ICL and fine-tuning, we seek an approach that is capable of:

- *Instruction following*: follows human-engineered instructions to achieve high quality predictions;

- *Chain-of-thought reasoning*: produces intermediate text that helps the model toward correct predictions;

- *Parameter tuning*: refines its internal representation to align with a moderate to large number of supervised training examples; and

- *Data scaling*: provides high quality models with data scales from 100 to 1000's of examples.

**Model stacking via CoT-augmented Tuning**    We begin with the observation that chain-of-thought prompting is typically *not* supervised, but rather induced via carefully-written instructions. Motivated by this, we fuse two models for learning and inference: a *big* model $\beta$ with all the most powerful emergent capabilities of LLMs, and a *tunable* model $\tau$ whose size can be flexibly chosen depending on the capacity needs of the task of interest. We assign the responsibility of chain-of-thought inference to $\beta$ and then provide its textual predictions $\hat{y}_\beta$ to the tunable model; it can then learn how to best use these inputs (e.g. chain-of-thought explanations) based on how useful they are with regard to predicting the supervised outputs. The parameters $\theta_\beta$ remain fixed as we do not have nor require any directly supervised data for its sub-task.

**Instruction-augmented Tuning**    Crafting a good instruction prompt is known to be essential to high-quality ICL performance, and so we naturally include instructions $I_\beta$ to generate reasoning and explanations as a first step. Although instructions are typically not used for smaller tunable model $I_\tau$, we observe that instructions have the potential to benefit tuning as well. We speculate that instructions help better align a task's inputs with the distribution seen during pre-training, allowing the model to not only converge faster but also make fewer parameter updates. This, in turn, avoids the risk of catastrophic forgetting associated with excessive parameter updates. Therefore, FIAT also provides separate instructions $I_\tau$ for the tunable model.[6]

**Pervasive Instruction-tuned Models**    Already, instruction-tuned models have become the standard for ICL; we use such models as $\theta_\beta$ in all of our experiments. However, given FIAT's use of Instruction-augmented Tuning, we also depart from the common practice of fine-tuning starting from models pre-trained primarily on span corruption objectives and instead initialize with instruction-tuned checkpoint (Longpre et al., 2023). This makes optimization easier since the model is already expecting instructions; this can be especially beneficial in limited training data scenarios.

**Parameter-efficient Tuning**    So far, we have added chain-of-thought reasoning, instruction following in tuning, and instruction-tuned initialization to FIAT's design, all of which move the pre-tuning model and the task definition toward each other in terms of increasing the probability of the desired output. We hypothesize that parameter-efficient tuning is a particularly good fit for optimizing $\theta_\tau$ in FIAT over the training data, because large changes to the model parameters $\theta_\tau$ should not be

---

[6]In FIAT, instructions can be viewed as serving purpose analogous to a Bayesian prior in earlier statistical learning methods: They allow encoding human knowledge into the learning procedure alongside supervised data that empirically estimates parameters. However, textual instructions are a far more natural way of doing this than the hyperparameters of a Dirichlet.

| **Algorithm 1:** Model building with FIAT | **Algorithm 2:** Inference with FIAT |
|---|---|
| **Input:** $\theta_\beta, \theta_\tau, \mathcal{D}$ | **Input:** $x, I_\beta, I_\tau, \theta_\beta, \theta'_\tau$ |
| **Output:** $\theta'_\tau, I_\beta, I_\tau$ | **Output:** $y$ |
| // Write reasoning instructions & select exemplars. | // Generate expansions, explanations, reasoning. |
| $I_\beta = \text{PROMPTENGINEERING}(\mathcal{D},\ \theta_\beta)$ | $\hat{y}_\beta = \arg\max_y P(y\|x; \theta_\beta, I_\beta)$ |
| // Write tuning instructions, based on large model. | // Infer final output using tuned model. |
| $I_\tau = \text{PROMPTENGINEERING}(\mathcal{D},\ \theta_\beta)$ | $y = \arg\max_y P(y\|x, \hat{y}_\beta; \theta'_\tau, I_\tau)$ |
| // Initialize parameter-efficient tuning. | |
| $\theta_\tau^{\text{PEFT}} \leftarrow \text{INIT}(\theta_\tau)$ | |
| // Iterate over examples or batches of data. | |
| **for** $x, y \in \mathcal{D}$ **do** | |
|    // Generate expansions, explanations, reasoning. | |
|    $\hat{y}_\beta = \arg\max_y P(y\|x; \theta_\beta, I_\beta)$ | |
|    // Optimize using parameter-efficient update. | |
|    $g_\tau = \nabla_{\text{PEFT}} P(y\|x, \hat{y}_\beta; \theta_\tau, \theta_\tau^{\text{PEFT}}, I_\tau)$ | |
|    $\theta_\tau^{\text{PEFT}} \leftarrow \text{UPDATE}(\theta_\tau^{\text{PEFT}}, g_\tau)$ | |
| **end** | |
| // Apply PEFT updates to final tuned model. | |
| $\theta'_\tau \leftarrow \theta_\tau \oplus \theta_\tau^{\text{PEFT}}$ | |

Figure 2: Model building and inference with FIAT. **Left:** Model building with FIAT begins with interactive prompt engineering of the instructions $I$. $I_\beta$ specifies how to perform reasoning using few-shot exemplars on $\theta_\beta$—i.e. behaviors for which we have no large-scale annotations, while $I_\tau$ specifies guidance to the tuned model $\theta_\tau$ for using the generated reasoning and input to produce a final output. Both $\theta_\beta$ and $\theta_\tau$ are instruction-tuned models and only $\theta_\tau$ is updated during training via parameter-efficient tuning. **Right:** Inference with FIAT is very simple, requiring only: (1) a call to the large generative model using the fixed pre-trained parameters $\theta_\beta$ and the reasoning instructions $I_\beta$; and (2) a call to the tuned model $\theta_\tau$ along with the associated task instructions $I_\tau$.

necessary given a good initialization.[7] Formalizing all the above modifications, we arrive at the final formulation of FIAT used for fine-tuning and inference in Alg. 1 and Alg. 2.

## 3 EXPERIMENTS

**Datasets** One of our primary objectives in selecting datasets that naturally cover a broad variety of training data sizes. We consider tasks ranging from classification to exercising a model's ability to generate short answers, and we include a large number and variety of languages to evaluate the generality of the method.

First, we use XOR-ATTRIQA (Muller et al., 2023), a classification task where model is asked to predict whether the provided answer to the question is supported by the given passage context, which includes 5 languages with 262 examples total. We refer to this as the $\mathcal{O}(100)$ data scenario.

We also study FIAT's behavior on the Cross-lingual QA task of XTREME-UP (Ruder et al., 2023). This data is an expansion of the XOR QA[8] dataset (Asai et al., 2020), a cross-lingual variant of the TyDi QA (Clark et al., 2020) dataset. This task asks a model to predict the correct English answer span given a non-English question and an English answer passage; this task also includes the possibility that the passage does not contain a correct answer, making it more challenging. Cross-lingual QA is a particularly important task for languages that have very little answer content as it enables providing answers to questions that would otherwise be unanswerable using only in-language content. We provide results on two focus sets. First, we use the subset of 20 Indic languages in XTREME-UP Cross-lingual QA where each language has about 300 examples, to allow for studying a scenario with

---

[7]In FIAT, we use LoRA (Hu et al., 2021) to parameterize the tuning procedure because it does not induce additional inference cost. Future work should consider other methods such as soft prompt tuning (Lester et al., 2021).

[8]XOR QA stands for cross-lingual open-retrieval question answering; note the difference between XOR QA and XOR-ATTRIQA.

| $\theta_\tau$ | $\theta_\beta$ | Method | XOR-ATTRIQA $\mathcal{O}(100)$ Acc / AUC-PR | XTREME-UP Cross-lingual QA (Indic) $\mathcal{O}(1000)$ F1 | XTREME-UP Cross-lingual QA (Full) $\mathcal{O}(10000)$ F1 |
|---|---|---|---|---|---|
| —— | L | ICL | 78.6 / ——[†] | 68.9 | 69.2 |
| XS | —— | Fine-tune | 90.5 / 52.1 | 63.5 | 75.5 |
|  | L | FIAT | 94.0 / 78.1 | 73.6 | 77.8 |
| S | —— | Fine-tune | 90.6 / 54.5 | 67.1 | 77.8 |
|  | L | FIAT | 93.9 / 77.5 | 77.3 | 79.3 |
| *Gain over best baseline* | | | +3.5 / +26.0 (vs S fine-tune) | +8.4 (vs ICL) | +1.5 (vs S fine-tune) |

Table 2: Overall results of FIAT and typical baselines. While we provide improvements with regard to the best baseline, we also point out that the best baseline often differs between ICL and fine-tuning, especially at smaller model sizes; this leaves practitioners to empirically determine the best course of action. [†]AUC-PR is not computed for the ICL because outputs are text-only.

moderate data; we refer to this as the $\mathcal{O}(1000)$ data scenario. We also study the full XTREME-UP Cross-lingual QA task which has 22,500 examples across 27 languages where the 5 high-resource languages have more than 2500 examples each; we refer to this as the $\mathcal{O}(10,000)$ data scenario.[9] Together, these tasks allow us to test our methods on three different data size scenarios from small 100's to over training 20,000 examples. Details of the languages and the dataset size can be found in App. A.1.

**Models** We use PaLM-2 (Google et al., 2023) as our base model, and we experiment with instruction-tuned models using the FLAN mixture (Chung et al., 2022). We use PaLM-2 L as $\mathcal{M}_\beta$ and we use PaLM-2 XS and S for $\mathcal{M}_\tau$.

**Baselines** We compare to both ICL and fine-tuning baselines. For ICL, we use PaLM-2 L with chain-of-thought reasoning (Wei et al., 2022b). We include 4 few-shot exemplars with hand-written chain-of-thought explanations in English for *each* of the 5 languages in the XOR-ATTRIQA Attribution task.[10] for a total of 20 exemplars. However, for XTREME-UP cross-lingual QA, it was not feasible to hand-engineer prompts for each of the 27 languages. Therefore, we hand-write 4 chain-of-thought explanations based on Bengali exemplars,[11] and use the same ICL examples for all 20 languages.

## 3.1 RESULTS

We present the performance of the baselines (ICL and fine-tuning) and our FIAT framework for all three data settings in Table 2. We show the average scores across all languages in each dataset for simplicity, and we provide the result for each language in App. A.2. Looking at the baselines, we find that few-shot ICL using PaLM-2 L model is quite competitive without any additional model tuning, but still lags behind PaLM-2 S fine-tuned on a relatively small amount of task data. However, we find that the best baseline differs between ICL and fine-tuning PaLM-2 XS across different tasks and data size settings. If one were choosing between just ICL or fine-tuning, this inconsistency makes it difficult to determine the best course of action without empirical comparisons. On the other hand, FIAT offers the best performance by combining the strengths of both ICL and fine-tuning.

## 4 ABLATIONS AND ANALYSIS

In this section, we study the effect of individual design decisions within FIAT and present the results in Table 3, and drawing conclusions from them below. In the end, we find that while certain design

---

[9]We report the average result on the under-represented languages, following the recommendations of the XTREME-UP benchmark.

[10]During manual prompt engineering, we used Google Translate to assist with explanation annotation.

[11]Note that while the exemplars have Bengali questions, we instruct the model to carry out its reasoning in English.

| $\theta_\tau$ | $\theta_\beta$ | Method | XOR-ATTRIQA O(100) Acc / AUC-PR | XTREME-UP Cross-lingual QA: Indics O(1000) F1 | XTREME-UP Cross-lingual QA: Full O(10000) F1 |
|---|---|---|---|---|---|
| —— | L | Few-shot ICL | 78.6 / —— | 68.9 | 69.2 |
| XS | L | FIAT | 94.0 / 78.1 | 73.6 | 77.8 |
| | —— | w/o CoT-augmentated tuning | 94.0 / 80.3 | 70.7 | 76.0 |
| | —— | w/o Instruction-augmented tuning | 93.5 / 72.4 | 69.8 | 76.4 |
| | —— | w/o Parameter-efficient tuning | 93.7 / 69.8 | 67.8 | 75.8 |
| | —— | w/o Instruction-tuned base model | 90.5 / 52.1 | 63.5 | 75.5 |
| S | L | FIAT | 93.9 / 77.5 | 77.3 | 79.3 |
| | —— | w/o CoT-augmentated tuning | 94.7 / 80.7 | 76.7 | 79.8 |
| | —— | w/o Instruction-augmented tuning | 94.1 / 71.6 | 75.3 | 79.1 |
| | —— | w/o Parameter-efficient tuning | 94.7 / 76.2 | 72.3 | 78.5 |
| | —— | w/o Instruction-tuned base model | 90.6 / 54.5 | 67.1 | 77.8 |

Table 3: Ablations showing the contribution of each modification within the FIAT recipe; each removal is cumulative with the one above. We observe that each modification tends to make a substantial positive impact on at least one scenario. The bottom line in each block is equivalent to traditional fine-tuning.

choices tend to have a larger effect on some settings than others, each tends to have substantial contributions in some area, and together the overall modeling recipe is very effective as a whole.

**Instructed-tuned base models improve final quality of fine-tuned models.** The instruction-tuned Flan XS model improves over the base model on all datasets, especially on XOR-ATTRIQA and XTREME-UP Cross-lingual QA Indic, where the total amount of task data is around $O(100)$ to $O(1000)$. This indicates that instruction-tuned models are not only beneficial for ICL, but can also be beneficial for fine-tuning on limited data (Longpre et al., 2023). However, the advantage of instruction-tuned model on XTREME-UP Cross-lingual QA decreases from the Indic ($O(1000)$ training examples) to Full ($O(10000)$ training examples), indicating that instruction-tuned model is less helpful when the fine-tuning dataset is large.

**Instruction-augmented Tuning generally leads to significant improvements.** Adding an appropriate prompted format to the task data is generally beneficial for all tasks. This result indicates that prompt engineering is not only helpful for direct few-shot ICL, but also has a positive impact on model fine-tuning. Prompted tuning is especially helpful for XOR-ATTRIQA and XTREME-UP Cross-lingual QA Indic, where the amount of task data is very limited. This is because the prompt format aligns the distribution of downstream task closer to the model pretraining distribution, which allows the pretrained model to generalize to the downstream task with a small amount of task examples.

**CoT-augmented Tuning is helpful for most tasks.** Our CoT-augmented Tuning can lead to large improvement for XTREME-UP Cross-lingual QA Indic task. Surprisingly, it does not help XOR-ATTRIQA, which is contradictory to findings from prior works which show that explanations can be especially helpful for classification tasks (Hsieh et al., 2023; Zhou et al., 2023). We hypothesize that this is because the model already performs quite well on XOR-ATTRIQA without having access to the explanations (over 90 percent accuracy) and this task may be reaching its saturation point.

**CoT-augmented Tuning is even more helpful for tasks and languages with lower performance.** We analyze the relationship between the gains brought by CoT-augmented Tuning on the XTREME-UP Cross-lingual QA tasks. Figure 3 shows the improvement in F1 score of different languages versus a baseline model's F1 score that lacks CoT-augmented Tuning. We can see that there is an inverse relationship between the benefit of CoT-augmented Tuning and the baseline model score, indicating that CoT is more beneficial for harder tasks or languages where the model could not perform well without the help of the CoT augmentation. This means that while we see meaningful gains in aggregate, for individual languages (or, more generally, individual tasks and use cases), CoT can have an out-sized impact on quality.

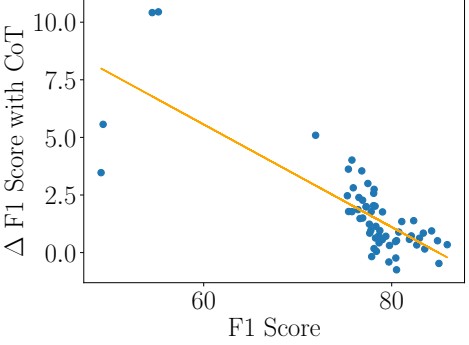

Figure 3: Gains in F1 on XTREME-UP Cross-lingual QA with CoT-augmented Tuning. The lower performing languages tend to benefit more from CoT augmentation.

| Method | F1 | Gains |
|---|---|---|
| Baseline | 70.7 | —— |
| Distilled CoT (Hsieh et al., 2023) | 72.5 | + 1.8 |
| Our CoT-augmented Tuning | 73.6 | + 2.9 |

Figure 4: Performance on XTREME-UP Cross-lingual QA Indic compared to the baseline without CoT. Our CoT-augmented Tuning method significantly outperforms previous methods on distilling CoT.

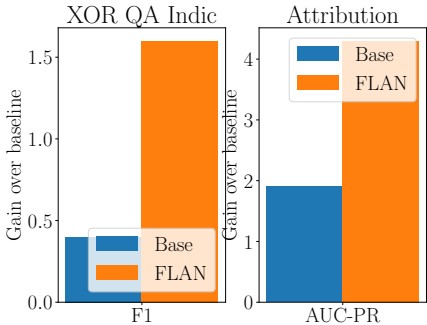

Figure 5: The validation F1 score throughout training on XTREME-UP Cross-lingual QA for methods with and without Instruction-augmented Tuning. Instruction-augmented Tuning out-performs baseline and it has much better performance at step 0, before any model optimization.

Figure 6: Improvement with Instruction-augmented Tuning for the model with and without instruction-tuning. Instruction-augmented Tuning is generally helpful for both types of models, and it tends to be more beneficial for instruction-tuned models

**CoT-augmented Tuning leads to better quality than CoT distillation.** Recent work proposed distilled CoT, which uses the explanation as a multitask output target, so that the model does not need to generate additional explanations at test time (Hsieh et al., 2023). Here we compare the performance of these two different ways of using the CoT explanations and list the performance on cross-lingual QA tasks in Figure 4. Despite incurring higher inference cost, our CoT augmentation method further out-performs the distilled CoT by a large margin on the harder XTREME-UP Cross-lingual QA Indic task. In general, we view distillation as an orthogonal technique to FIAT, which is aimed at efficiency over quality.

**Adding instructions to tuning helps from beginning to end.** In Figure 5, we plot the training curves of Flan PaLM-2 S model with and without Instruction-augmented Tuning. We can see that adding instructions to tuning leads to much better performance at step 0, before any model optimization. This indicates that adding the instructions to the task data *during fine-tuning*[12] can significantly improve the *zero-shot* performance of the model, probably because it makes the task

---

[12]Note we use the term **instruction-augmented tuning** to differentiate from the separate concepts of **instruction-tuned base models**, which creates base models that are better able to follow instructions of specific tasks later, and **prompt tuning**, which learns soft prompt embeddings.

data more similar to the data used in the instruction tuning stage. Importantly, this also implies that the model parameters don't need to move as far away from their starting point in order to achieve the same level of quality, reducing the risk of catastrophic forgetting. However, the model does not only reach the same level of quality with less steps, but also manages to exceed the quality of a model without instructions.

**Instruction-augmented Tuning helps more with an instruction-tuned base model.** We compare the effect of prompted tuning on models with and without instruction tuning. Figure 6 shows that prompted tuning generally brings improvements for both the base model without instruction tuning and the Flan model with instruction tuning, while the gains on the instruction-tuned Flan model tend to be slightly larger and more consistent. This is likely because the data format we used for prompted tuning (task instructions followed by the input) is more similar to the Flan data mixture used for instruction tuning.

## 5    RELATED WORK

**Instruction Tuning**    Instruction-tuned models (Wei et al., 2021; Longpre et al., 2023) often have better performance for few-shot ICL tasks than base language models since they are already primed to following instructions due to being fine-tuned on a diverse set of tasks. Using instruction-tuned models is a key component of FIAT.

**In-Context Learning**    In in-context learning, the parameters of the LLM remain fixed and a prompt containing a few examples along with reasoning steps is used to prime the model for solving similar tasks  (Nye et al., 2021; Wei et al., 2022b). In-context learning works best for large language models. FIAT uses this capability of large language models, along with fine-tuning, to power small language models in the low-data regime.

**Knowledge Transfer from Larger to Smaller LLMs**    A popular prior method for transferring knowledge from large models to smaller ones is model distillation (Hinton et al., 2015), where the outputs of a larger model are used as a training signal for a smaller one. Other approaches include using the larger language model to generate data and then using this data to train smaller models. More recently, the latter has approach has been extended to generate reasoning steps which are provided as fine-tuning data for the smaller language model (Magister et al., 2022; Huang et al., 2022; Li et al., 2022; Ho et al., 2023; Hsieh et al., 2023; Fu et al., 2023; Zhu et al., 2023; Li et al., 2023).

**Under-represented Languages**    Most work that trains large language model and uses them for downstream tasks focus on English or the collection of 100 or so languages where there are large, easily available corpora (ImaniGooghari et al., 2023). Tail languages have often been ignored by language technologies due to lack of available corpora (Nayak & Joshi, 2022). Recent works has focused on tail languages outside of these head languages (Bapna et al., 2022; Ruder et al., 2023). In this work, we make the low-data regime the focus of our efforts, which is especially useful for tail languages.

**Fine-tuning smaller LLMs**    While fine-tuning with prompts has been studied for encoders pre-trained with masked language modeling objectives (Scao & Rush, 2021), we show that it is also important to fine-tuning generative language models. For example, some works show that fine-tuning a smaller language model is a more competitive and efficient method for practical low-data learning problems than few-shot ICL (Asai et al., 2023; Ruder et al., 2023). Agrawal et al. (2022) propose to synthetic QA data generated from very large LLM to improve the performance of a smaller model.

## 6    CONCLUSION

We have presented FIAT, a method that fuses the ICL and fine-tuning learning paradigms and leads to improved model predictions across a variety of data scenarios, ranging from 100–10,000 training examples. We hope FIAT provides a practical way of harnessing the full potential of LLMs without needing to make a hard choice between learning paradigms.

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

| Split | bn | fi | ja | ru | te |
|---|---|---|---|---|---|
| Train | 40 | 66 | 20 | 84 | 52 |
| Validation | 218 | 150 | 578 | 136 | 174 |
| Test | 2822 | 1318 | 1908 | 1268 | 2146 |

Table 4: Dataset size for XOR-ATTRIQA.

| Split | as | bho | brx | gbm | gom | gu | hi | hne | kn | mai | ml | mni | mr | mwr | or | pa | ps | sa | ta | ur | ar | bn | fi | ja | ko | ru | te |
|---|---|---|---|---|---|---|---|---|---|---|---|---|---|---|---|---|---|---|---|---|---|---|---|---|---|---|---|
| Train | 323 | 326 | 326 | 326 | 326 | 326 | 326 | 326 | 326 | 326 | 326 | 326 | 326 | 326 | 326 | 326 | 326 | 326 | 326 | 326 | 3159 | 377 | 2467 | 2926 | 3327 | 2560 | 373 |
| Validation | 356 | 358 | 357 | 365 | 365 | 371 | 519 | 372 | 373 | 369 | 373 | 380 | 385 | 386 | 386 | 385 | 384 | 385 | 384 | 387 | 941 | 618 | 978 | 727 | 861 | 731 | 468 |
| Test | 633 | 631 | 633 | 634 | 629 | 630 | 1049 | 629 | 631 | 635 | 629 | 628 | 633 | 632 | 632 | 624 | 633 | 630 | 630 | 634 | 582 | 397 | 606 | 471 | 548 | 448 | 333 |

Table 5: Dataset size for XTREME-UP Cross-lingual QA.

# A APPENDIX

## A.1 LIST OF LANGUAGES FOR EACH TASK

We provide the number of training, validation, and test examples for each task in Table 4 and Table 5.

## A.2 LANGUAGE-WISE BREAKDOWN OF THE RESULTS

We provide the performance for each language in Table 6, Table 7, and Table 8.

| $\mathcal{M}_\tau$ | $\mathcal{M}_\beta$ | Method | bn | fi | ja | ru | te |
|---|---|---|---|---|---|---|---|
| | | | | | Acc / AUC-PR | | |
| —- | L | Few-shot ICL | 85.9 / —- | 78.5 / —- | 85.4 / —- | 84.5 / —- | 58.9 / —- |
| | L | FIAT | 92.6 / 81.1 | 91.0 / 85.3 | 96.3 / 66.5 | 94.8 / 84.9 | 95.3 / 72.5 |
| | —- | w/o CoT-Augmented Tuning | 92.5 / 84.7 | 91.8 / 85.8 | 96.2 / 70.3 | 94.6 / 84.1 | 95.0 / 76.6 |
| XS | —- | w/o Instruction-Augmented Tuning | 91.7 / 74.1 | 91.2 / 81.4 | 95.9 / 53.5 | 93.8 / 77.4 | 94.8 / 75.4 |
| | —- | w/o Parameter-efficient Tuning | 92.6 / 73.9 | 92.0 / 76.7 | 95.0 / 55.8 | 94.2 / 74.1 | 94.7 / 68.6 |
| | —- | w/o Instruction-tuned base model | 89.4 / 65.6 | 88.9 / 65.9 | 94.3 / 42.1 | 90.1 / 58.6 | 89.7 / 28.2 |
| | L | FIAT | 92.3 / 81.3 | 92.1 / 84.0 | 96.2 / 62.4 | 94.6 / 84.9 | 94.0 / 93.9 |
| | —- | w/o CoT-Augmented Tuning | 93.0 / 84.3 | 94.4 / 81.2 | 95.5 / 58.8 | 98.8 / 87.4 | 95.3 / 78.4 |
| S | —- | w/o Instruction-Augmented Tuning | 93.1 / 75.6 | 92.7 / 82.9 | 95.0 / 51.3 | 94.6 / 78.1 | 95.2 / 70.1 |
| | —- | w/o Parameter-efficient Tuning | 92.7 / 76.2 | 93.2 / 83.6 | 96.3 / 59.0 | 95.1 / 83.3 | 96.5 / 78.8 |
| | —- | w/o Instruction-tuned base model | 90.9 / 66.3 | 88.6 / 67.7 | 93.2 / 41.0 | 89.7 / 57.5 | 90.3 / 40.2 |

Table 6: Results on each language for XOR-ATTRIQA.

| $\mathcal{M}_\tau$ | $\mathcal{M}_\beta$ | Method | as | bho | brx | gbm | gom | gu | hi | hne | kn | mai | ml | mni | mr | mwr | or | pa | ps | sa | ta | ur |
|---|---|---|---|---|---|---|---|---|---|---|---|---|---|---|---|---|---|---|---|---|---|---|
| | | | | | | | | | | | | | | F1 | | | | | | | | |
| — | L | Few-shot ICL | 72.5 | 61.8 | 43.0 | 60.3 | 72.3 | 70.6 | 61.5 | 70.8 | 72.9 | 73.3 | 72.2 | 57.1 | 71.5 | 69.5 | 71.4 | 73.7 | 70.6 | 72.6 | 71.5 | 69.4 |
| XS | L | FIAT | 75.9 | 73.9 | 47.2 | 72.7 | 76.1 | 76.1 | 79.3 | 76.2 | 76.6 | 75.5 | 76.3 | 61.1 | 75.4 | 73.3 | 76.0 | 75.6 | 76.6 | 77.4 | 75.4 | 73.3 |
| | — | w/o CoT-Augmented Tuning | 73.2 | 73.0 | 40.7 | 68.8 | 71.3 | 76.1 | 79.0 | 72.3 | 74.0 | 71.4 | 76.7 | 48.8 | 73.3 | 72.3 | 71.6 | 74.6 | 72.2 | 74.9 | 75.0 | 74.7 |
| | — | w/o Instruction-Augmented Tuning | 73.2 | 71.5 | 39.1 | 67.8 | 71.7 | 73.7 | 78.5 | 70.3 | 74.0 | 71.2 | 74.7 | 50.1 | 73.9 | 71.4 | 70.9 | 72.2 | 72.8 | 71.8 | 74.5 | 72.48 |
| | — | w/o Parameter-efficient Tuning | 70.7 | 69.5 | 49.2 | 65.7 | 70.7 | 80.5 | 67.4 | 69.9 | 69.7 | 70.9 | 51.6 | 70.0 | 67.8 | 66.8 | 69.5 | 69.7 | 68.7 | 70.9 | 69.8 | 67.8 |
| | — | w/o Instruction-tuned base model | 65.6 | 64.7 | 49.3 | 60.3 | 62.6 | 65.7 | 76.9 | 63.2 | 65.2 | 63.7 | 65.4 | 52.8 | 64.2 | 63.5 | 63.8 | 65.8 | 64.3 | 63.7 | 65.4 | 64.4 |
| S | L | FIAT | 80.2 | 77.8 | 52.2 | 77.2 | 78.3 | 80.6 | 82.2 | 79.5 | 79.7 | 78.8 | 79.8 | 64.5 | 79.4 | 77.4 | 79.4 | 80.7 | 80.0 | 80.4 | 79.8 | 78.0 |
| | — | w/o CoT-augmented Tuning | 79.1 | 78.4 | 50.3 | 75.6 | 78.7 | 79.9 | 84.6 | 77.8 | 79.2 | 78.3 | 79.2 | 62.4 | 77.8 | 77.7 | 79.6 | 79.2 | 78.8 | 79.9 | 80.1 | 78.0 |
| | — | w/o Instruction-Augmented Tuning | 78.8 | 77.6 | 47.7 | 75.1 | 76.1 | 79.1 | 82.8 | 76.3 | 78.4 | 78.0 | 78.4 | 58.0 | 78.1 | 76.0 | 79.3 | 78.1 | 77.0 | 78.2 | 78.0 | 77.2 |
| | — | w/o Parameter-efficient Tuning | 74.3 | 71.2 | 50.6 | 71.7 | 72.7 | 74.6 | 81.8 | 72.7 | 75.1 | 74.1 | 74.9 | 61.9 | 73.9 | 72.1 | 75.8 | 75.5 | 73.5 | 72.6 | 73.6 | 73.5 |
| | — | w/o Instruction-tuned base model | 68.8 | 68.2 | 46.1 | 66.5 | 67.5 | 69.0 | 79.4 | 68.8 | 69.4 | 68.3 | 69.4 | 53.5 | 68.4 | 67.1 | 69.2 | 68.4 | 69.4 | 67.3 | 70.0 | 68.0 |

Table 7: Results on each language for XTREME-UP Cross-lingual QA Indic.

| $\mathcal{M}_\tau$ | $\mathcal{M}_\beta$ | Method | as | bho | brx | gbm | gom | gu | hi | hne | kn | mai | ml | mni | mr | mwr | or | pa | ps | sa | ta | ur | ar | bn | fi | ja | ko | ru | te |
|---|---|---|---|---|---|---|---|---|---|---|---|---|---|---|---|---|---|---|---|---|---|---|---|---|---|---|---|---|---|
| | | | | | | | | | | | | | | | F1 | | | | | | | | | | | | | | |
| — | L | Few-shot ICL | 72.5 | 61.8 | 43.0 | 60.3 | 72.3 | 70.6 | 61.5 | 70.8 | 72.9 | 73.3 | 72.2 | 57.1 | 71.5 | 69.5 | 71.4 | 73.7 | 70.6 | 72.6 | 71.5 | 69.4 | 66.0 | 75.2 | 65.5 | 60.3 | 61.2 | 66.9 | 68.7 |
| XS | L | FIAT | 80.1 | 80.4 | 52.6 | 77.0 | 78.9 | 80.7 | 85.2 | 80.5 | 80.8 | 79.0 | 79.6 | 65.6 | 79.6 | 78.7 | 79.8 | 79.1 | 80.1 | 79.5 | 79.8 | 78.3 | 83.7 | 84.6 | 82.8 | 83.7 | 86.3 | 81.6 | 82.4 |
| | — | w/o CoT-augmented Tuning | 79.8 | 76.8 | 49.1 | 71.9 | 76.5 | 78.1 | 84.2 | 77.5 | 79.0 | 75.4 | 79.0 | 55.2 | 77.8 | 75.9 | 75.8 | 78.7 | 78.1 | 78.3 | 80.5 | 78.1 | 83.5 | 85.0 | 82.1 | 82.3 | 85.9 | 80.8 | 81.1 |
| | — | w/o Instruction-augmented Tuning | 78.8 | 77.8 | 49.2 | 72.8 | 77.0 | 78.7 | 83.9 | 76.8 | 80.1 | 76.1 | 80.4 | 58.3 | 78.7 | 76.2 | 77.1 | 78.6 | 76.8 | 79.1 | 79.4 | 79.4 | 84.5 | 84.6 | 81.5 | 82.6 | 87.0 | 81.7 | 80.8 |
| | — | w/o Parameter-efficient Tuning | 78.3 | 75.6 | 55.4 | 74.7 | 75.0 | 78.0 | 84.9 | 76.5 | 78.9 | 77.3 | 78.8 | 61.9 | 77.8 | 77.3 | 75.9 | 78.4 | 76.9 | 76.6 | 79.8 | 77.8 | 84.3 | 83.5 | 81.9 | 83.2 | 88.1 | 80.4 | 81.3 |
| | — | w/o Instruction-tuned base model | 76.9 | 76.4 | 56.6 | 73.1 | 74.2 | 76.8 | 84.7 | 75.4 | 77.9 | 75.5 | 78.1 | 62.8 | 77.5 | 74.3 | 74.7 | 77.5 | 76.5 | 75.3 | 77.5 | 75.8 | 82.4 | 84.2 | 81.2 | 82.8 | 88.1 | 80.4 | 80.3 |
| S | L | FIAT | 81.6 | 80.5 | 51.9 | 78.3 | 80.2 | 82.3 | 85.8 | 81.2 | 82.4 | 82.1 | 81.5 | 67.0 | 82.1 | 80.2 | 81.6 | 80.9 | 81.5 | 82.2 | 82.3 | 79.5 | 82.5 | 86.2 | 82.0 | 83.7 | 87.1 | 83.3 | 86.2 |
| | — | w/o CoT-augmented Tuning | 82.8 | 80.5 | 49.9 | 78.0 | 80.0 | 83.4 | 85.9 | 80.4 | 82.7 | 80.5 | 83.7 | 64.9 | 81.5 | 80.2 | 82.0 | 82.0 | 83.0 | 82.4 | 80.0 | 84.2 | 86.6 | 81.9 | 82.4 | 87.0 | 83.9 | 84.3 | 80.6 |
| | — | w/o Instruction-augmented Tuning | 81.3 | 80.0 | 51.2 | 78.3 | 78.4 | 82.0 | 85.7 | 80.5 | 81.2 | 80.3 | 81.8 | 64.8 | 81.0 | 79.7 | 81.2 | 80.5 | 80.7 | 80.5 | 81.6 | 79.4 | 82.8 | 85.7 | 83.3 | 83.8 | 86.4 | 84.1 | 84.0 |
| | — | w/o Parameter-efficient Tuning | 79.5 | 77.5 | 61.5 | 77.3 | 78.3 | 80.1 | 85.3 | 79.0 | 79.9 | 79.0 | 80.5 | 68.9 | 79.0 | 78.4 | 79.8 | 78.8 | 78.7 | 78.9 | 80.5 | 78.3 | 83.3 | 85.1 | 84.1 | 84.9 | 89.2 | 85.7 | 82.4 |
| | — | w/o Instruction-tuned base model | 79.5 | 77.4 | 55.4 | 75.6 | 79.1 | 79.9 | 85.5 | 77.5 | 80.7 | 78.5 | 80.3 | 63.4 | 79.5 | 77.8 | 78.8 | 78.6 | 78.7 | 78.8 | 80.7 | 77.7 | 81.9 | 85.8 | 84.0 | 85.0 | 88.8 | 91.9 | 82.1 |

Table 8: Results on each language for XTREME-UP Cross-lingual QA All.

