# OpenReview forum: "FIAT: Fusing learning paradigms with Instruction-Accelerated Tuning"
_ICLR.cc/2024/Conference — ICLR 2024 Conference Withdrawn Submission_

### Official Review · Reviewer_qVcA · 2023-10-30

**Soundness:** 3 good
**Presentation:** 2 fair
**Contribution:** 3 good
**Rating:** 5
**Confidence:** 4

**Summary:**

ICL and fine-tuning of the model have their individual pros and cons. These drawbacks often compel professionals to select a paradigm arbitrarily or conduct expensive experiments on these distinct methods to determine the optimal approach. The paper suggests that these two model learning paradigms complement each other. The proposed approach, FIAT, integrates ICL in very large models and parameter tuning in moderately-sized LLM, amalgamating the shared techniques from each paradigm.

**Strengths:**

- The paper clearly explains the strengths and weaknesses of both ICL and parameter tuning. It makes it to grasp the idea and motivation more easily.
- It is easy to read.
- The ablation study for techniques included in the paradigm was conducted.

**Weaknesses:**

-  While some are empirically proven in the Ablation study section, when describing the need for various techniques included in the algorithm, it seems somewhat heuristic, giving a feeling of subjective choice. For example, I don't quite understand the part that describes the reason for including instructions in the tunable model in Section 2.
- The explanation for why fine-tuning performs exceptionally well despite the very small training dataset size of XOR-ATTRIQA is insufficient. Given the apparent contradiction with the pros and cons of previous ICL and fine-tuning, there seems to be a need for an explanation.
- The explanation for the process of prompt engineering is insufficient.
- The types of the three presented tasks seem too similar. I believe that validating the algorithm with different types of tasks could further solidify it.

**Questions:**

- Could you explain in more detail about the part, "We speculate that instructions help better align a task’s inputs with the distribution seen during pre-training" in "Instruction-augmented Tuning" paragraph of Section 2?
- How do you select exemplars when you do prompt engineering?
- Could you explain in more detail the reason why the algorithm uses the same ICL examples based on Bengali exemplars?
- Was the experiment conducted for each task only once?
- Could you please explain the reason why the performance seems to be better when CoT-augmented tuning is removed in the XOR-ATTRIQA task in Table 3?
- Was the baseline Fine-tuning presented without instructions? If so, could you provide the results of Fine-tuning along with instructions?

---

### Official Review · Reviewer_47ne · 2023-10-30

**Soundness:** 3 good
**Presentation:** 3 good
**Contribution:** 3 good
**Rating:** 6
**Confidence:** 3

**Summary:**

This paper integrates the In-Context Learning of large LM and parameter-efficient tuning of medium LM and proposes a novel paradigm: FIAT. The FIAT takes the chain-of-thought inference from a large model as an auxiliary input to the medium model for parameter-efficient fine-tuning. In the experiments, FIAT outperforms the baselines across multiple benchmark tasks.

**Strengths:**

1. This paper reviews the strengths and weaknesses of two popular paradigms for adapting LLM to specific downstream tasks, i.e., ICL and PEFT. The proposed FIAT combines the advantages of ICL and PEFT. On the one hand, it can leverage the knowledge from the most capable LLMs, and on the other hand, it utilizes PEFT from the modestly-sized LLM with training data from the downstream task. The experiments demonstrate that FIAT indeed benefits from these two aspects and achieves performance improvement.
2. FIAT seems to be somewhat similar to the previous neural network distillation: obtaining soft labels from a large model to improve the performance of a small model. In this work, the large model provides CoT explanations as few-shot demonstrations for the small model. This utilization method appears to be clever and consistent with the idea of providing demonstrations for downstream tasks as ICL does. The authors also show through experiments that utilizing CoT explanations in the manner of ICL is superior to using them as multitask outputs.

**Weaknesses:**

The authors claim that the proposed FIAT provides a way of harnessing the full potential of LLMs without making a choice between In-Context Learning and Fine-Tuning. Personally, this might be a bit of an overclaim for the contribution of this work. Although this work combines the in-context inference of large models and the parameter-efficient fine-tuning of medium models, it primarily holds true for limited computation costs. Under limited compute cost, for large models like the PaLM L model, we can only use ICL to adapt the model to specific downstream tasks and can not perform PEFT. On the other hand, from the perspective of combining the ICL and PEFT paradigms, it is worth noting that the models used in this paper are not the same. A more appropriate expression could be: combining the ICL of a large model and the PEFT of a medium model.

Miscellaneous:
1. In the knowledge transfer paragraph of Section 5: “More recently, the latter has approach has been extended”.
2. There is overlap in the subfigures of Figure 6.

**Questions:**

1. As claimed in this paper, the context window limited the utilization of more additional training examples. Adding CoT explanations in ICL may reduce the length of instruction that can be provided. In practice, how to control the length of CoT explanation, that is, how to take the balance between the $I_\tau$ and $\hat{y}_\beta$?
2. Taking more additional data into the PEFT could help the LLM adapt to the specific downstream task. This is relatively easy to understand. What is the role of adding the CoT explanations in the Instruction for performance improvement? Besides its high-level help of taking the capacity from a larger LLM, does it have any further impact? Does FIAT potentially assume that the LLM ($\theta_\beta$) performs well for the current task in order to obtain meaningful explanations instead of providing misleading ones?

---

### Official Review · Reviewer_oPRp · 2023-10-31

**Soundness:** 2 fair
**Presentation:** 1 poor
**Contribution:** 1 poor
**Rating:** 1
**Confidence:** 4

**Summary:**

The paper proposes FIAT, a learning paradigm designed to overcome trade-offs in in-context learning (ICL) and full fine-tuning on LLM at different scales. FIAT exploits hand-engineered instruction prompts that elicit chain-of-thought reasoning while also using the generated reasoning steps and input prompts to tune a relatively small LLM with parameter efficient tuning methods on the downstream task. The effectiveness of the method is tested on multilingual tasks.

**Strengths:**

The proposed approach is designed to reduce the computational cost of adapting LLMs on downstream tasks.

**Weaknesses:**

1. **(Novelty)**. The proposed method provides little novel contribution, the basic idea is to use an expert LLM to generate data to improve downstream models has already been explored in very similar scenarios. For example, [1] an expert LLM to generate possible continuations that are then filtered (to improve quality) and used for subsequent fine-tuning to get the final downstream model. Retrieval Augmented Generation methods (e.g. [2]) follow the same idea but, rather than generating new samples form an expert LLM, they leverage an external source of information to fill the model context before generation. Furthermore, [3] follows a similar trajectory, where, continuations of language only GPT-4 are directly used as supervised data for instruction tuning of Visual Language Models.
Overall, the idea proposed in this paper can be seen as a form of automatic data augmentation/synthetic data generation, as such it is very close to previously proposed methods both in NLP and in the intersection between Vision and Language.
2. **(Clarity)** The paper is quite hard to read and often unfocused jumping between different methods while adding a lot of repetitions. See for example Section 2.3. And, it is quite hard to find a clear scientific hypothesis that the authors want to test/validate.
3. **(Soundness)** While augmenting the context of LLM with tokens from other expert (strong models) or even humans is not new and has been proven to work well in practice multiple times. I am a bit puzzled by some claims in the paper. LLMs are know to often hallucinate irrelevant content or just provide answers containing many repetitions (especially when greedy decoding is used as in the first stage of generation proposed here). So, how did authors make sure that the CoT explanation is of high quality? And in the worst case, i.e. when some bad explanations are given, how does the smaller LLM recover from the misleading context? In the current manuscript no intuitions are given nor algorithmic solutions are attempted to ameliorate this problem.


[1] Yizhong W. et al., "SELF-INSTRUCT: Aligning Language Models with Self-Generated Instructions"

[2] Aleksandra P. et al.,"Retrieval-Augmented Generation for Knowledge-Intensive NLP Tasks"

[3] Haotian L. et al., "LLaVA: Large Language and Vision Assistant"

**Questions:**

1. Can the authors add some of the continuations of the first LLM in the manuscript? That would help the reader better delineating how many tokens are typically required to get the reported results.
2.  Can you further comment the results in Table 3 regarding the S model \theta_\tau? Why is FIAT performing so bad?

---

### Official Review · Reviewer_nxJx · 2023-11-02

**Soundness:** 3 good
**Presentation:** 2 fair
**Contribution:** 1 poor
**Rating:** 3
**Confidence:** 4

**Summary:**

This paper proposed a framework to combine several learning paradigms for large language models and achieves state-of-the-art performance on several tasks.

**Strengths:**

The paper studies an important problem that how to better combine different learning paradigms for better performance with less resources.

**Weaknesses:**

1. Although the studied problem is interesting and important, the authors should dive deeper into how to better combine them instead of simply adding them altogether. For example, how to better choose $I_\beta$ to achieve better training of $\tau$? How to write the $I_\tau$? There are many unexplored problems in this paper.
2. Presentation of this paper is unclear, especially in Figure 2. What does the $y$ refer to in $\hat{y}_{\beta} = \text{argmax}_y P(y| x;\theta, I)$?
$y$ is defined as a target output sampled from the training data $D$ on the previous line but used as a variable on this line.
3. As mentioned in point 1, the combination explored in this work is too straight-forward for venues such as ICLR and doesn't introduce new knowledge and findings to the community. Combining multiple partial training methods altogether must have some improvement as long as there is no overfitting.

**Questions:**

Please see the weakness part.